# Comparison of Efficacy of 2% Chlorhexidine Gluconate–Alcohol and 10% Povidone-Iodine–Alcohol against Catheter-Related Bloodstream Infections and Bacterial Colonization at Central Venous Catheter Insertion Sites: A Prospective, Single-Center, Open-Label, Crossover Study

**DOI:** 10.3390/jcm11082242

**Published:** 2022-04-17

**Authors:** Ming-Ru Lin, Po-Jui Chang, Ping-Chih Hsu, Chun-Sui Lin, Cheng-Hsun Chiu, Chih-Jung Chen

**Affiliations:** 1Division of Pediatric Infectious Diseases, Department of Pediatrics, Chang Gung Memorial Hospital, Taoyuan 333, Taiwan; b9102095@cgmh.org.tw (M.-R.L.); chchiu@cgmh.org.tw (C.-H.C.); 2School of Medicine, College of Medicine, Chang Gung University, Taoyuan 333, Taiwan; r5478@cgmh.org.tw (P.-J.C.); 8902049@cgmh.org.tw (P.-C.H.); 3Department of Thoracic Medicine, Chang Gung Memorial Hospital, Taipei 105, Taiwan; 4Division of Thoracic Medicine, Linkou Chang Gung Memorial Hospital, Taoyuan 333, Taiwan; 5Infection Control Committee, Chang Gung Memorial Hospital, Taoyuan 333, Taiwan; i22061@cgmh.org.tw; 6Molecular Infectious Diseases Research Center, Chang Gung Memorial Hospital, Taoyuan 333, Taiwan

**Keywords:** chlorhexidine gluconate–alcohol, povidone-iodine–alcohol, catheter-related bloodstream infection, colonization

## Abstract

An effective antiseptic agent is an essential component of a central venous catheter (CVC) care bundle, to protect against catheter-related bloodstream infections (CRBSIs). We conducted a trial to compare the incidences of CRBSI and the growth of insertion site flora in patients with CVC using 2% chlorhexidine gluconate–alcohol (CHG) or 10% povidone-iodine–alcohol (PVI) in the CVC care bundle. Patients who were admitted to two medical intensive care units (ICUs) and had CVC placement for >48 h were enrolled. Using a two-way crossover design with two six-month interventions, the ICUs were assigned to use either CHG or PVI in their care bundles. A total of 446 catheters in 390 subjects were enrolled in the study. The detection rate of flora was greater in the PVI group on both day 7 (26.6% versus 6.3%, *p* < 0.001) and day 14 (43.2% versus 15.8%, *p* < 0.001). The incidence rate of CRBSI was higher in the PVI group compared to the CHG group (2.15 vs. 0 events per 1000-catheter-days, *p* = 0.001), although the significance was lost in the multivariate analysis. In conclusion, 2% CHG was superior to 10% PVI in the CVC care bundle in terms of the inhibition of skin flora growth at CVC insertion sites and was potentially associated with lower incidence rates of CRBSI.

## 1. Introduction

Catheter-related bloodstream infections (CRBSIs) are common and serious infections in healthcare settings [1,2,3]. The prevention of CRBSIs can improve patient outcomes while greatly reducing the length of their hospital stays and medical costs [4]. A prevention bundle that includes multimodal intervention has been demonstrated to be effective against CRBSIs. Nevertheless, the goal of further reducing the prevalence of CRBSIs is considered meaningful and achievable [5]. 

Chlorhexidine and povidone-iodine have been proven to be effective when used in the repair of oral wounds, as they control local infections with a low risk of systemic allergy and toxicity [6]. Furthermore, chlorhexidine is able to suppress microorganisms in the oral cavity, meaning it is effective in the improvement of marginal bone loss after implant insertion in dental procedures [7], the treatment of gingivitis and periodontitis [8], and in reducing aspiration pneumonia in patients who underwent major surgery [9].

The skin surrounding insertion sites is believed to be the main source of infection when catheters are placed for shorter durations [10]. Therefore, along with bundle intervention, the use of an optimal skin antiseptic agent in catheter insertion and maintenance is crucial in the prevention of CRBSIs. Several studies have demonstrated that chlorhexidine gluconate (CHG) solution provides better effectiveness in decreasing the incidence rate of CRBSIs compared to 10% povidone-iodine (PVI) solution, either with peripheral venous catheters, central venous catheters (CVCs), or arterial catheters [11,12,13,14,15]. However, these studies had several methodological drawbacks that meant the general application of their findings was limited. Furthermore, few studies have compared the efficacy of the two antiseptic agents regarding the inhibition of the growth of skin flora over catheter insertion sites.

According to the current guidelines of the Centers for Disease Control and Prevention in the US, chlorhexidine solution at a concentration of >0.5% with alcohol is recommended during catheter insertion and dressing change for the prevention of CRBSIs [16]. The solution combines the immediate microbiocidal activity of alcohol and the persistent activity of chlorhexidine on skin [17,18,19]. However, CHG is much more expensive than PVI in Taiwan, and it has been debated that the routine use of CHG as an antiseptic agent in bundle interventions was more cost effective than the use of PVI. To evaluate the direct effects of these solutions on the inhibition of the growth of skin flora at CVC insertion sites and the potential effects they have in the prevention of CRBSIs, we conducted a prospective study to compare the performance of 2% CHG and 10% PVI in CVC care bundles in two medical intensive care units (ICUs).

## 2. Materials and Methods

### 2.1. Study Design

From 1 October 2016 to 31 October 2017, we conducted a prospective, open-label, crossover trial in two medical ICUs in Chang Gung Memorial Hospital in Taiwan (Clinical trials registration number 201600826B0C101). In 2015, the incidence rates of CRBSI in the two ICUs were 4.5 per 1000-catheter-days and 5.5 per 1000-catheter-days, respectively, and 10% PVI–alcohol was used as the antiseptic agent in the CVC care bundle. At the start of the study, all of the procedures used in the CVC bundle care were standardized and taught to all of the healthcare personnel in the two ICUs, which included barrier precaution during CVC insertion, skin preparation, the method of changing dressings and surveillance culture methods. The two ICUs possessed bed-sizes of 25 and 15. They both aimed to admit patients with disease related to internal medicine, and mainly related to the pulmonary system. 

### 2.2. Ethics Statement

The study protocol was reviewed and approved by the Research Ethics Committee of the Chang Gung Memorial Hospital in 2016 (No. 201600826B0C101). Written informed consent was obtained from all of the participants and their legal representatives. 

### 2.3. Subjects

Patients who were admitted to the two ICUs with CVC placements that had been inserted for longer than 48 h were enrolled. Patients were excluded from this study if the CVC insertion was carried out before ICU admission, if they had bacteremia before CVC insertion, if they had known intolerance, allergic history or adverse events to any trial drug, and if they died within 48 h after admission to the ICUs.

### 2.4. Intervention and Clinical Procedures

Except for the antiseptic agents, all of the CVCs were inserted and maintained using identical procedures in the two ICUs included in the study. Using a two-way crossover design, the ICUs were assigned to use 2% CHG–alcohol or 10% PVI–alcohol in their respective CVC care bundles. There were two intervention periods of six months separated by a washout period of one month (Appendix A).

The types of catheter and insertion sites used in this study were selected at the discretion of individual physicians. Patients who received silver antiseptic or antimicrobial-impregnated catheters were excluded. All of the physicians who participated in the study were required to follow CBC recommendations [16] by using maximal barrier precautions. Before catheter insertion, the site was prepared using the allocated antiseptic solution and allowed to dry according to standardized protocols, that is, moving back and forth and then being allowed to dry for 30 s once for 2% CHG–alcohol and moving in circular motions and then being allowed to dry for 30 s, which was repeated three times for 10% PVI–alcohol. After catheter insertion, sterile, semi-permeable, transparent or gauze dressings (if local oozing or exudate were observed) were placed at the insertion site. The dressings were changed every 3 days for gauze dressings, every 7 days for sterile transparent dressings, or sooner if they became soiled or wet. The same assigned antiseptic solution was used at each dressing change. The use of dressings that contained antiseptic or local antimicrobial ointment was not allowed.

Physicians and nurses evaluated the indication of CVC insertion and inspected the insertion site every 24 h. Catheters were removed if there was no more indication for use, any evidence of local infection (e.g., swelling, erythema, pain, or pus discharge) or if a CRBSI was suspected. CRBSIs were suspected in those who experienced fever (body temperature ≥ 38 °C), hypothermia (body temperature ≤ 36 °C), chillness or sudden shock (systolic blood pressure < 90 mmHg).

### 2.5. Bacteriological Methods

Paired blood samples for aerobic culture were drawn simultaneously from a peripheral venous site and via a catheter hub to determine the differential time to positivity when a CRBSI was suspected. Meanwhile, 5 cm of a distal CVC tip was sent for semi-quantitative culture after catheter removal. A surveillance culture over the CVC insertion site was carried out to evaluate the growth of skin flora on day 7 and day 14 after catheter insertion by pressing a sterilized cotton swab circularly on the skin for 10 s (Appendix A). All of the samples were aerobically cultured at 35~37 °C for 24 h to 7 days. The identification of the isolated bacteria was carried out using the matrix-assisted laser desorption ionization–time of flight (MLDI-TOF) technique. Antimicrobial susceptibility was determined following the recommendation of the Clinical and Laboratory Standards Institution.

### 2.6. Definition of Catheter-Related Bloodstream Infection

We defined a catheter-related bloodstream infection (CRBSI) as a combination of the following: fever or hypothermia; positive blood culture drawn from peripheral veins 48 h before or after catheter removal; isolation of the same organism (same species and same antimicrobial susceptibility) from CVC tip culture; no clear source of bacteremia other than the catheter. 

### 2.7. Outcome

The primary endpoint was the incidence of CRBSI per 1000-catheter-days at the time of catheter removal, and the secondary endpoint was the detection of the growth of skin flora at the insertion site on day 7 and day 14 after CVC insertion.

### 2.8. Statistical Analysis

We used the chi-square test or Fisher’s exact test to compare categorical variables. Non-categorical variables were compared using a one-way independent analysis of variance. Kaplan–Meier analysis was used to evaluate catheter survival during CRBSI. The primary population analyzed in this study was the full analysis set (FAS) group, while patients without catheter colonization data were excluded from the intention-to-treat (ITT) group. Data analyses were performed using SPSS software version 20.0 (SPSS Inc., Chicago, IL, USA). A *p*-value < 0.05 was considered statistically significant.

## 3. Results

### 3.1. Subject Enrollment

During October 2016 and October 2017, 520 CVCs were used in 464 patients in the two ICUs under study. Of these, 501 CVCs in 445 subjects who provided consent were enrolled to this study. After the exclusion of subjects who died, left ICUs or had CVCs removed within 48 h of admission, 446 CVCs in 390 subjects qualified for the FAS review, including 256 in the 2% CHG group and 190 in the 10% PVI group. Surveillance cultures over the CVC insertion sites were available for 263 CVCs in 237 subjects on day 7 and 113 in 102 subjects on day 14, which were considered the ITT group (Figure 1).

### 3.2. Demographics and Clinical Characteristics

The demographics and baseline clinical characteristics of the participants are displayed in Table 1. Underlying autoimmune diseases were more commonly identified in the subjects in the 10% PVI group than in those in the 2% CHG group. Most clinical characteristics, including the total hospital days, ICU days and the antibiotic treatment before CVC insertion, were similar among the two groups, except that the subjects in the CHG group tended to have higher APACHE II scores. The distributions of the CVC insertion sites were also similar between the two groups.

### 3.3. Catheter-Related Bloodstream Infection and Microbiologic Features

The incidence of CRBSI was 0 and 1.59 events per 1000-catheter-days in the CHG group and PVI group (*p* = 0.002 by log-rank test of Kaplan–Meier curve), respectively, in FAS. However, the significance disappeared after controlling the underlying conditions and the APACHE II score using Cox-regression (hazard ratio (HR): 0.998, 95% confidence interval (CI) 0.887–1.123, *p* = 0.971). In ITT analysis, the incidence of CRBSI was 0 and 2.14 events per 1000-catheter-days in the CHG and the PVI group, respectively (*p* = 0.001 by log-rank test of Kaplan–Meier curve). The difference was not significant in the analysis with the Cox proportional hazard model (HR: 0.09, 95%CI: 0–5.187, *p* = 0.146). The pathogens accountable for the CRBSIs are displayed in Table 2. 

### 3.4. Surveillance Culture of the Catheter Insertion Site

Surveillance cultures were obtained from 263 CVCs in 237 subjects. Of these, 59 (22.4%) participants had bacterial colonization over the insertion sites on the 7th or 14th day after the implementation of CVC. The overall colonization rate was significantly higher in participants in the PVI group than those in the CHG group (38/105, 36.2% versus 20/158, 12.7%, *p* < 0.001). The positive yields and the microbiological features of the skin flora in the two timepoints of surveillance are displayed in Table 3. The detection rates were consistently lower in the 2%CHG group, both on day 7 or day 14. The participants in the PVI group tended to be more commonly colonized with Gram-negative bacilli, fungi and multiple micro-organisms. The usage of PVI in the CVC care bundle was the only factor associated with the increased incidence of flora detection in the participants (odds ratio: 4.08, CI: 2.21–7.53, *p* < 0.001, Table 4).

## 4. Discussion

The results of this study demonstrated that, compared to 10% PVI, 2% CHG can significantly reduce the prevalence of skin flora colonization over CVC insertion sites and can potentially decrease the incidence rates of CRBSIs in patients in ICUs. Previous studies have shown that a multimodal intervention for implanted central catheters was effective in reducing the incidence rates of CRBSIs in adult ICUs [5,20]. Consistent with previous reports, the number of CRBSIs per 1000-catheter-days in the ICUs included in this study decreased from 4.5–5.5 events before the study to 1.59 events in the PVI group in this study. This finding suggested that, in addition to the use of antiseptic agents, continuing the education of healthcare personnel in terms of enforcing the correct methods of catheter insertion and maintenance in CVC bundle care is crucial in the prevention of CRBSIs in ICUs. 

In 2015, Mimoz O et al. demonstrated that the usage of 2% CHG–alcohol as a skin antiseptic provided a five-fold decrease in the incidence of short-term CRBSIs compared with 5% PVI–alcohol using a well-designed, open-label, randomized, controlled trial [21]. Consistent with this report, we found a trend that showed 2% CHG–alcohol was superior to 10% PVI–alcohol in terms of its preventive effects against CRBSIs in ICUs. CRBSIs exclusively occurred in the PVI group, and none of the patients in the CHG group had an infection. The loss of the significance of the antiseptic agents in the multivariate analysis might be due to the small number of CRBSI events in this study. A larger-scale study is required to accurately estimate the differences in CRBSI incidences in patients treated with different antiseptic agents.

The skin that surrounds insertion sites is believed to be the main source of bloodstream infection during shorter-duration catheter insertion, while the hub or connector is thought to play a more important role when catheters are inserted for longer timeframes [10]. Flora colonization at insertion sites can be reasonably considered a surrogate endpoint to the efficacy against CRBSIs in the evaluation of the usefulness of topical antiseptic solutions. The results from our study demonstrated a significant decrease in skin colonization at insertion sites when 2% CHG–alcohol antiseptic was used, either at D7 or D14 after catheter insertion. The inhibition effect was most prominent for Gram-negative organisms and fungi. It was noteworthy that the superiority of 2% CHG–alcohol over 10% PVI–alcohol was not affected by the type of admission (community versus hospital associates), the disease severity of patients, the duration of a patient’s hospital stay before CVC insertion or the insertion sites. A meta-analysis published in 2016 also concluded that CHG-based formulations might reduce the rates of catheter colonization [22]. Meanwhile, Yasuda et al. showed that the use of 10% PVI as an antiseptic with catheters in situ over 72 h increased the risk of catheter colonization compared to either 0.5% CHG–alcohol or 1% CHG [23].

There were several limitations to this study. Firstly, it was an open-label trial, and the participants were not randomized to receive either antiseptic agent. However, a detailed protocol, including each step of bundle care, surveillance culture techniques and handling of the antiseptic solution was well defined and distributed to each ICU. The crossover study design was also expected to largely decrease the influences of different ICU settings and healthcare staff. Furthermore, the microbiologists involved in the study were blinded to the solution used. We believe that the results are not severely biased. Secondly, the baseline data of the clinical characteristics between the two study groups were not completely balanced. However, an attempt to control the potential confounding effects of the covariates was made by using two robust statistic methods including multivariate and Cox-regression analysis. We believe that the effect of both antiseptic agents on the inhibition of the growth of skin flora was unambiguously estimated.

## 5. Conclusions

In conclusion, 2% CHG was superior to 10% PVI in the CVC care bundle in terms of the inhibition of skin flora growth at CVC insertion sites, and it was potentially associated with lower incidence rates of CRBSIs. Therefore, 2% CHG–alcohol is preferred as a component of CVC bundle intervention in critically ill patients. The effect of 2% CHG on reducing incidence rates of CRBSI was not proven to be significant in our study, which might have been caused by the low incidence of CRBSIs and the small size of our study group. Thus, further studies that concern monitoring reductions in CRBSI rates after the routine use of 2% CHG as an antiseptic solution in our hospital should be considered. Furthermore, the effect of 2% CHG as an antiseptic agent on catheters inserted for longer periods of time, such as peripherally inserted central catheters in patients who require peripheral nutrition, could be another interesting research topic.

## Figures and Tables

**Figure 1 jcm-11-02242-f001:**
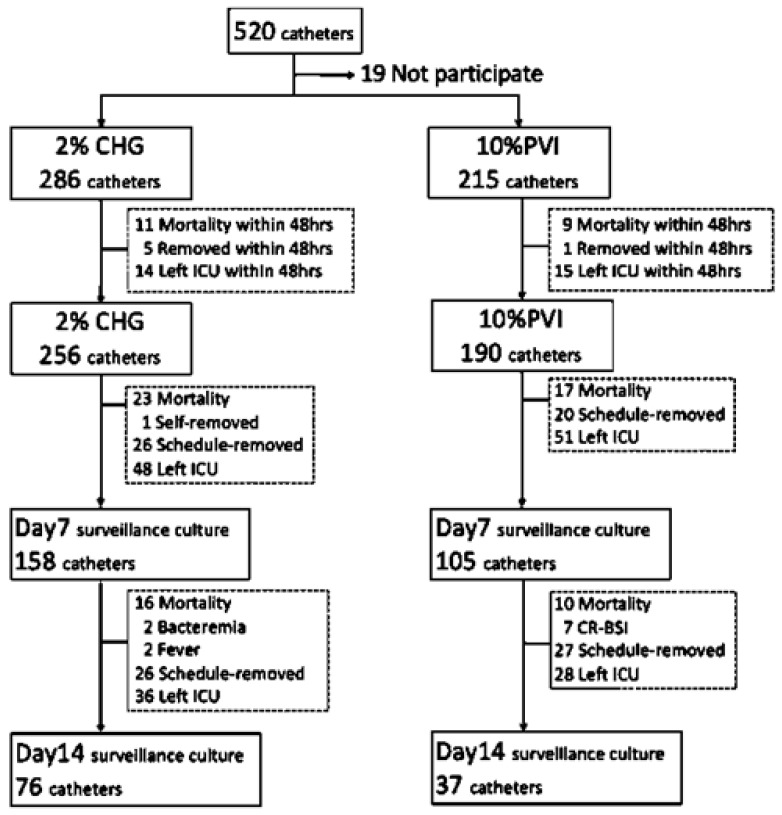
Flow chart of case enrollment. Abbreviations: CHG, 2% chlorhexidine gluconate–alcohol; PVI, 10% povidone-iodine–alcohol; HIV, human immunodeficiency virus; CVC, central venous catheter; ICU, intensive care unit; CRBSI, catheter-related bloodstream infection.

**Table 1 jcm-11-02242-t001:** Demographics and clinical characteristics of study participants.

Characteristics	Full Analysis Set			Intention-to-Treat		
	CHG *n* (%)	PVI *n* (%)	*p*	CHG *n* (%)	PVI *n* (%)	*p*
Case number	256	190		158	105	
Demographics						
Age, mean (SD), year	66.59 ± 15.1	67.35 ± 16.0	0.605	67.96 ± 13.32	67.1 ± 15.79	0.631
Gender, male	189 (73.8)	120 (63.2)	0.016	122 (77.2)	71 (67.6)	0.085
From community	186 (72.7)	136 (71.6)	0.802	116 (73.4)	71 (67.6)	0.31
Clinical condition						
APACHE II, mean (SD)	26.45 ± 7.4	23.47 ± 8.35	0.005	27.3 ± 6.85	25.1 ± 7.7	0.084
Chronic underlying disease	231 (90.2)	181 (95.3)	0.048	147 (93.0)	100 (95.2)	0.465
HIV	0	1 (0.5)	0.426	0	1 (1.0)	0.399
Malignancy	84 (32.8)	57 (30.0)	0.528	53 (33.5)	38 (36.2)	0.659
Autoimmune disease	3 (1.2)	9 (4.7)	0.034	0	3 (2.9)	0.063
Hospital days before CVC	16.11 ± 20.91	14.71 ± 19.22	0.469	17.53 ± 20.1	15.24 ± 19.89	0.364
ICU days before CVC	6.72 ± 10.67	5.98 ± 9.03	0.429	7.82 ± 11.51	6.3 ± 9.77	0.252
CVC site						
Internal jugular vein	68 (26.6)	42 (22.1)	0.279	49 (31.0)	25 (23.8)	0.203
Subclavian vein	11 (4.3)	10 (5.3)	0.63	5 (3.2)	6 (5.7)	0.355
Femoral vein	177 (69.1)	138 (72.6)	0.42	104 (65.8)	74 (70.5)	0.431
Antibiotics before CVC	245 (95.7)	183 (96.3)	0.745	154 (97.5)	100 (95.2)	0.33
Infection sites for antibiotics use						
Pulmonary	202 (78.9)	146 (76.8)	0.6	133 (84.2)	76 (72.4)	0.02
Central nervous system	0	0	NA	0	0	NA
Cardiovascular	3 (1.2)	2 (1.0)	>0.99	3 (3.9)	2 (1.9)	>0.99
Abdominal	7 (2.7)	1 (0.5)	0.146	3 (1.9)	1 (0.95)	0.652
Urinary tract	9 (3.5)	6 (3.2)	0.84	7 (4.5)	0	0.044
Soft tissue/bone	5 (1.9)	5 (2.6)	0.75	2 (1.3)	4 (3.8)	0.22
Others	23 (9.0)	23 (12.1)	0.28	11 (7.1)	17 (16.2)	0.017

Abbreviations: CHG, 2% chlorhexidine gluconate–alcohol; PVI, 10% povidone-iodine–alcohol; HIV, human immunodeficiency virus; CVC, central venous catheter; ICU, intensive care unit; APACHE II, Acute Physiology and Chronic Health Evaluation score; NA, not applicable.

**Table 2 jcm-11-02242-t002:** Incidence and microbiologic features of the catheter-related bloodstream infections.

Characteristics	Full Analysis Set			Intention-to-Treat		
	CHG *n* (%)	PVI *n* (%)	*p*	CHG *n* (%)	PVI *n* (%)	*p* ^e^
Catheter numbers	256	190		158	105	
Incidence (1000-catheter-day)	0	1.59	0.002	0	2.14	0.001
Flora						
Gram-positive ^a^	0	2	NA	0	2	NA
Gram-negative ^b^	0	3	NA	0	3	NA
CONS ^c^	0	2	NA	0	2	NA
Fungus ^d^	0	1	NA	0	1	NA
Drug-resistant	0	2	NA	0	2	NA

Abbreviations: CHG, 2% chlorhexidine gluconate–alcohol; PVI, 10% povidone-iodine–alcohol; CRBSI, catheter-related bloodstream infection; CONS, coagulase-negative Staphylococcus; NA, not applicable. ^a^ Including 1 oxacillin-resistant *Staphylococcus aureus*; 1 vancomycin-resistant *Enterococcus faecalis*; ^b^ Including 3 *Burkholderia cepacia* complex; ^c^ Including 1 *Staphylococcus capitis*; 1 *Staphylococcus haemolyticus*; ^d^
*Candida albican*; ^e^
*p* value was calculated using log-rank test of Kaplan–Meier curve.

**Table 3 jcm-11-02242-t003:** Incidences and microbiologic features of microbial colonization at the central catheter insertion sites.

Characteristics	Day 7			Day 14		
	CHG *n* (%)	PVI *n* (%)	*p*	CHG *n* (%)	PVI *n* (%)	*p*
Catheter numbers	158	105		76	37	
Flora detected	10 (6.3)	28 (26.6)	<0.001	12 (15.8)	16 (43.2)	0.002
Flora						
Gram-positive	1 (0.6)	5 (4.8)	0.039	3 (3.9)	2 (5.4)	>0.99
Gram-negative	6 (3.8)	17 (16.2)	<0.001	5 (6.6)	7 (18.9)	0.056
CONS	4 (2.5)	15 (14.3)	<0.001	4 (5.3)	9 (24.3)	0.004
Fungus	1 (0.6)	4 (3.8)	0.084	0	3 (8.1)	0.033
Multiple pathogen	2 (1.3)	12 (11.4)	<0.001	0	4 (10.8)	0.01
Drug-resistant	6 (3.8)	8 (7.6)	0.176	5 (6.6)	4 (10.8)	0.47

Abbreviations: CHG, 2% chlorhexidine gluconate–alcohol; PVI, 10% povidone-iodine–alcohol; CONS, coagulase-negative Staphylococcus.

**Table 4 jcm-11-02242-t004:** Factors associated with flora colonization at the CVC insertion sites.

Characteristics	Colonized (N = 59) *n* (%)	Non-Colonized (N = 204) *n* (%)	*p*-Value
Age, mean ± SD (year)	67.47 ± 14.75	67.65 ± 14.25	0.935
From hospital associate	18 (30.5)	58 (28.4)	0.757
Chronic underlying disease	56 (94.9)	191 (93.6)	0.716
Immune-compromised	20 (33.9)	75 (36.8)	0.689
APACHE II score	26.0 ± 7.40	26.49 ± 7.26	0.739
Hospital days before CVC	19.47 ± 24.27	15.79 ± 18.58	0.284
ICU days before CVC	7.31 ± 11.68	7.19 ± 10.64	0.994
Antibiotics treatment	57 (96.6)	197 (96.6)	0.944
CVC site			
Internal jugular vein	19 (32.2)	55 (27.0)	0.431
Subclavian vein	2 (3.4)	9 (4.4)	>0.99
Femoral vein	38 (64.4)	140 (68.6)	0.543
TPN supplement	1 (1.7)	2 (1.0)	0.536
Antiseptic via PVI	39 (66.1)	66 (32.4)	<0.001

Abbreviations: PVI, 10% povidone-iodine–alcohol; CVC, central venous catheter; ICU, intensive care unit; SD, standard deviation. APACHE II, Acute Physiology and Chronic Health Evaluation score.

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
