# Peer review of "Comparison of Efficacy of 2% Chlorhexidine Gluconate–Alcohol and 10% Povidone-Iodine–Alcohol against Catheter-Related Bloodstream Infections and Bacterial Colonization at Central Venous Catheter Insertion Sites: A Prospective, Single-Center, Open-Label, Crossover Study"

_jcm, 2022, doi:10.3390/jcm11082242_

Round 1

Reviewer 1 Report

The authors perform an open label, crossover trial in 2 medical ICUs to try and describe if there is a difference between the CHG+alcohol and PVI+alcohol skin antisepsis for catheter related bloodstream infection rates.  They also attempted to explain the difference by studying skin colonization after antisepsis. While this difference or lack thereof would be an interesting finding to know, the authors do not explain their rational for performing the study.  It is stated that the CRBSI in both ICUs was relatively high at 4.5 and 5.5 per 1000 catheter days before the study began, so perhaps they wanted to assess the intervention of switching skin antisepsis.

I would like to ask several questions to help clarify some of the authors thoughts:

Introduction:

1) In line 40, “along with bundle care” – what is meant by bundle care? Perhaps you mean a bundled intervention as described in the studies cited?

2) Would consider adding the rationale for performing the study in this section, why were you all interested in this?

Materials and Methods:

  • Were both ICUs already performing the bundled care intervention for CVC placement with PVI prior to the study? If so then the PVI group is the control group correct? Or was there another intervention that you are trying to account for such as renewed education, bundle adherence tracking and reporting, or safety culture was addressed somehow? It is clear there was something else done in both groups since the CRBSI rate went down from 4.5 and 5.5 per 1000 catheter days to 1.59. Or is this the Hawthorne effect that lasted for a year long?
  • Were there any power calculations performed to try and determine the sample size needed to detect significant change in the primary outcome? Perhaps they were calculated based on the pre-trial CRBSI rate and then given the low number of CRBSIs during the trial period, this number of infections were lower than anticipated and thus perhaps affected your ability to detect a significant change in CRBSI?
  • Would like to see more information on the ICUs, such as bed-size, type of patients, mostly medical, cardiac, or any surgical overflow.
  • Would for the authors to further describe the bundled intervention performed during CVC insertion and show the actual differences in the groups pre trial and between the two intervention groups.
  • Ideally would also like to see a patient flow diagram with the flow of the trial over time including how the intervention group moved from one ICU to the other
  • Bacteriologic sampling, was the day 7 and day 14 skin sampling performed during the routine dressing change? Was it done after repeat skin antisepsis was performed on that occasion, were there other times the skin underwent antisepsis, such as if the dressing fell off or there were some bleeding?
  • Definition of the CRBSI, was this determined by a trained person evaluating both ICUs and both groups or were there separate people performing the ascertainment of whether there was an infection or not? Was there a way to delineate the CONS bloodstream contaminants from true infections? This definition seems somewhat outdated, meaning, we do not typically require the pathogen identified in the blood stream to be the same identified from the culture tip in order for it to be considered a CRBSI.  However the number of blood cultures that are positive matter depending on the pathogen grown. Did you all use any national guideline for this definition, if so, would cite that guidance.
  • Outcome – It seems each patient was analyzed according to the antisepsis assignment but was there also an analysis between the ICUs? Was there a difference in CRBSI or skin colonization rate over time during the trial?  After the cross-over were the two groups’ rates of CRBSI similar to before the cross-over?
  • Intention to Treat group designation is really just the FAS minus those patients without colonization data. This is not really what an intention to treat group is.  However, it would be helpful to see the intention to treat data, e.g. what proportion of patients in the CHG group actually received CHG antisepsis, do you have any bundle adherence data?
  • Kaplan-Meier analysis was mentioned in the Statistical analysis, but I did not see this in the results as a survival or time to event analysis, which would be very interesting. Also, multivariate regression was mentioned in the abstract and discussion but not described here.

Results:

  • Would be curious to see this table 1 for the ICUs unless they are exactly the same already.
  • CRBSI – as mentioned above, would need to see the pre-trial CRBSI rate in both ICUs and whether there was a significant difference between the ICUs during the trial. Ideally the analysis could be done for the primary outcome by treatment group and by the ICUs to make sure there were no implementation differences. 
  • Table 2 – what do you mean by “drug resistant”?
  • Table 3 – presumably the CONS were excluded from the Gram-positive group? Can you describe what you mean by drug-resistant?
  • Were the 3 Gram-negative CRBSIs all Burkolderia in different patients? Was this an outbreak

Discussion:

  • Would include some explaination of the difference in the CRBSI rate after the trial started and whether there was a trend over time in improvement. Another idea would be to perform an Interupted Time Series Analysis on the treatment groups to see if the intervention of CHG was associated with the reduction in CRBSI or if it was related to time and other intervnetions that were happening simultaneously. But this may be for another paper.

Author Response

We appreciated the constructive feedback received from both the Editor and Reviewers, and we have addressed each of your concerns in files.

Reviewer 2 Report

The article entitled “Comparison of efficacy of 2% chlorhexidine gluconate-alcohol and 10% povidone-iodine-alcohol against catheter-related bloodstream infections and bacterial colonization at central venous catheter insertion sites” aimed discuss the incidences of catheter-related bloodstream infections (CRBSI) and the composition of microbiological colonization in patients with CVC using 2% chlorhexidine gluconate-alcohol (CHG) or 10% povidone-iodine-alcohol (PVI).

The paper is in line with journal’s aim, moreover, Authors have well revised several issues; however, I ask authors to add some key concepts.

  • In the title of the article, authors should specify the type of study
  • In the introduction section, the authors should add previous findings on selected antimicrobial agents also on other body districts (e.g., CHX is an agent able to inhibit plaque formation and remains the safest and most effective antimicrobial agent used for the reduction of microorganisms in the oral cavity, please see and discuss DOI: 10.1155 / 2018/5326340)
  • It would be interesting to include a flow chart regarding the study design
  • The limits of the study should be included in the paper
  • Conclusions cannot be reduced to a sentence: you must improve them highlighting the limits and the future insights pointed out from this article.
  • The formatting of the references is not correct, please check the journal instructions for authors
  • Several moderate typos are present in the text, please, amend

Author Response

We appreciated the constructive feedback received from both the Editor and Reviewers, and we have addressed each of your concerns in file.
